# Tumour-Infiltrating Lymphocytes (TILs) and PD-L1 Expression Correlate with Lymph Node Metastasis, High-Grade Transformation and Shorter Metastasis-Free Survival in Patients with Acinic Cell Carcinoma (AciCC) of the Salivary Glands

**DOI:** 10.3390/cancers13050965

**Published:** 2021-02-25

**Authors:** Selina Hiss, Markus Eckstein, Patricia Segschneider, Konstantinos Mantsopoulos, Heinrich Iro, Arndt Hartmann, Abbas Agaimy, Florian Haller, Sarina K. Mueller

**Affiliations:** 1Institute of Pathology, University Hospital Erlangen, Friedrich-Alexander University Erlangen-Nürnberg (FAU), Krankenhausstr. 8-10, 91054 Erlangen, Germany; selina.hiss@uk-erlangen.de (S.H.); markus.eckstein@uk-erlangen.de (M.E.); patricia.segschneider@uk-erlangen.de (P.S.); arndt.hartmann@uk-erlangen.de (A.H.); abbas.agaimy@uk-erlangen.de (A.A.); 2Department of Otorhinolaryngology, Head & Neck Surgery, University Hospital Erlangen, Friedrich-Alexander University Erlangen-Nürnberg (FAU), Waldstrasse 1, 91054 Erlangen, Germany; Konstantinos.Mantsopoulos@uk-erlangen.de (K.M.); heinrich.iro@uk-erlangen.de (H.I.); sarina.mueller@uk-erlangen.de (S.K.M.)

**Keywords:** PD-L1, tumour-infiltrating lymphocytes, checkpoint inhibition, immuno-oncology, acinic cell carcinoma

## Abstract

**Simple Summary:**

Acinic cell carcinoma (AciCC) is a malignant neoplasm of the salivary glands. The assessment of tumour-infiltrating lymphocytes (TILs) and the evaluation of the expression of *Programmed Cell Death 1 Ligand 1* (PD-L1) can provide prognostic information and may be valuable in predicting response to immuno-oncological treatments. The aim of our retrospective study was to measure the number of TILs and PD-L1 expression in 36 AciCCs and to analyze their prognostic impact based on the correlation with clinico-pathological features. Increased numbers of TILs and a higher expression of PD-L1 were associated with tumour high-grade transformation and the presence of lymph node metastasis. Moreover, higher expression of PD-L1 correlated with a shorter period of metastasis-free survival. As a result, the correlation between TILs, PD-L1 expression, high-grade transformation and lymph node metastasis indicates a relevant interaction between tumour cells and immune cell infiltration and might constitute the basis for innovative immunological therapy attempts.

**Abstract:**

Objectives: The aim of this study was to assess the number of tumour-infiltrating lymphocytes (TILs) and the expression of *Programmed Cell Death 1 Ligand 1* (PD-L1) in Acinic Cell Carcinoma (AciCC) of the salivary glands, to enable a correlation with clinico-pathological features and to analyse their prognostic impact. Methods: This single centre retrospective study represents a cohort of 36 primary AciCCs with long-term clinical follow-up. Immunohistochemically defined immune cell subtypes, i.e., those expressing T-cell markers (CD3, CD4 and CD8) or a B-cell marker (CD20) were characterized on tumour tissue sections. The number of TILs was quantitatively evaluated using software for digital bioimage analysis (QuPath). PD-L1 expression on the tumour cells and on immune cells was assessed immunohistochemically employing established scoring criteria: tumour proportion score (TPS), Ventana immune cell score (IC-Score) and combined positive score (CPS). Results: Higher numbers of tumour-infiltrating T- and B-lymphocytes were significantly associated with high-grade transformation. Furthermore, higher counts of T-lymphocytes correlated with node-positive disease. There was a significant correlation between higher levels of PD-L1 expression and lymph node metastases as well as the occurrence of high-grade transformation. Moreover, PD-L1 CPS was associated with poor prognosis regarding metastasis-free survival (*p* = 0.049). Conclusions: The current study is the first to demonstrate an association between PD-L1 expression and lymph node metastases as well as grading in AciCCs. In conclusion, increased immune cell infiltration of T and B cells as well as higher levels of PD-L1 expression in AciCC in association with high-grade transformation, lymph node metastasis and unfavourable prognosis suggests a relevant interaction between tumour cells and immune cell infiltrates in a subset of AciCCs, and might represent a rationale for immune checkpoint inhibition.

## 1. Introduction

Acinic cell carcinoma (AciCC) constitutes the third most common malignant epithelial neoplasm of the salivary glands [1]. Compared to other salivary gland malignancies, patients with AciCC generally have a favourable prognosis with a 10-year relative survival of 88% [1,2]. However, local recurrences and distant metastases have been reported in up to 35% and 12% of patients, respectively, sometimes years or even decades after the initial treatment [1,3,4]. The main prognostic factors for survival are age, tumour size, lymph node metastasis, high-grade transformation and distant metastasis [1,3,4,5,6]. While enucleation and local excision of the tumour are obsolete due to an elevated risk of local recurrence [7,8], elective neck-dissection is considered only in those patients with locally advanced disease or with high-grade transformation of the tumour [9,10,11,12]. Notably, clinically occult regional lymph node metastases have been reported in up to 22% of patients with AciCC, and elective neck dissection for each patient with AciCC independent of other clinico-pathological parameters has been suggested by a recent study [5,12,13,14] High-grade transformation in AciCC has originally been defined as the presence of a dedifferentiated component with features of high-grade adenocarcinoma or poorly differentiated carcinoma next to a differentiated conventional low-grade AciCC component, however, cases fulfilling this definition are exceedingly rare [15,16,17]. Several histomorphological features including increased mitotic counts, nuclear pleomorphism, necrosis and infiltrative growth pattern correlate with reduced survival in patients with AciCC [3,5,18,19] but no uniform histological grading system for AciCC has been established yet [4,5].

The role of tumour-infiltrating lymphocytes (TILs) has seldom been explored in AciCC [20]. TILs play an important role in antigen-specific tumour immune response, and recent studies showed that a higher number of TILs is an unfavourable prognostic factor for survival in salivary gland cancers [21,22]. However, AciCCs constituted only minor subgroups in these cohorts with mixed histological cancer types, and the entity-specific prognostic role of TILs has been so far limited to adenoid cystic carcinoma [23,24]. Similar to the presence of TILs, the expression of *Programmed Cell Death 1 Ligand 1* (PD-L1) has been reported to be of unfavourable prognostic significance in cohorts of mixed salivary gland cancers [22,25,26,27], as well as in salivary duct cancer [13,28,29]. In general, expression of PD-L1 is correlated with tumour response to immune checkpoint inhibition, but the applied immunohistochemical antibodies, scoring systems and cut-offs for positivity vary greatly. In salivary gland cancers, response rates to therapies including immune checkpoint blockade have been limited, but partial responses in individual patients were observed [30,31,32,33,34]. Therefore, the objective of this study was the comprehensive analysis of TILs and PD-L1 expression in correlation to clinico-pathological parameters including long-term follow-up in a well-characterized single institutional cohort of AciCCs of the salivary glands.

## 2. Materials and Methods

### 2.1. Collection of Tissue Specimens

The study cohort comprised 36 patients with primary AciCC resected at the Department of Otorhinolaryngology, Head and Neck Surgery, University Hospital of the Friedrich-Alexander University Erlangen-Nürnberg between 2003 and 2019. All cases have been published previously, and diagnosis of AciCC and grading have been established by an experienced Head and Neck pathologist (A.A.) as described previously [35]. The presence of tertiary lymphoid structures was identified qualitatively on H&E slides if lymphoid structures with dense cellular aggregates resembling germinal centres were observed within the tumour area. The presence of true lymph node metastasis was critically revaluated; in cases with prominent lymphoid stroma, the presence of prominent pan keratin-positive reticulum cells within the lymphoid background was used as evidence of genuine lymph node and lack of such reticulum cells was considered consistent with reactive lymphoid stroma as proposed recently by Kurian et al. [36]. Clinico-pathological data including follow-up were collected from the clinical archive, with (i) recurrence-free survival defined as time period from date of surgery until the date of local recurrence, (ii) metastasis-free-survival defined as time period from date of surgery until the date of distant metastasis and (iii) disease-specific survival defined as time period from date of surgery until the date of death from tumour disease. The study had been approved by the ethic committee of the Medical Faculty of the Friedrich-Alexander University Erlangen-Nuremberg on 10 January 2018 (ethic code: 355_17 Bc). All patients gave written informed consent for the study, and the samples were used in accordance with ethical guidelines for the use of retrospective tissue samples provided by the local ethics committee of the Friedrich-Alexander University Erlangen-Nuremberg (ethics committee statements 24 January 2005 and 18 January 2012).

### 2.2. Immunohistochemical Analysis

A representative paraffin-embedded tissue block was identified, and immunohistochemistry was performed on freshly cut 4-μm thick tissue sections using a fully automated staining system (Ventana BenchMark Ultra immunostainer, Roche, Mannheim, Germany). After deparaffinisation in xylene, the sections were rehydrated with graded ethanol at room temperature. After incubation with the primary antibodies (Table 1), sections were washed and incubated with biotinylated secondary antibodies. Immunoreactions were visualized using the Ultraview DAB Detection Kit (Roche). Human tonsillar tissue and lymph node tissue were used as positive controls. PD-L1 expression was semi-quantitatively determined using established scores by two experienced pathologists (F.H. and M.E.). The Tumour Proportion Score (TPS) was defined as the percentage of viable tumour cells showing partial or complete membrane PD-L1 staining at any intensity, the Immune Cell Score (IC-Score) as the percentage of tumour area covered by PD-L1 positive immune cells, and the Combined Positive Score (CPS) as the number of PD-L1 staining cells (tumour cells, lymphocytes, macrophages), divided by the total number of viable tumour cells, multiplied by 100 [37]. Independent scoring revealed a good concordance between two pathologists for scoring of TPS (Pearson correlation coefficient *r* = 0.85), IC-Score (*r* = 0.74) and CPS (*r* = 0.72), and consensus assessment was accomplished in few cases with diverging initial results. Representative examples of two AciCCs (case 6, Figure 1 A–D and case 35, Figure 1 E–H) are shown in Figure 1.

### 2.3. Digital Scoring of Immunohistochemistry

Slides were digitized using a Pannoramic P250 Flash (3DHistech, Budapest, Hungary) slide scanner and processed in QuPath v.0.1.2 (Queen’s University, Belfast, Northern Ireland). Digital images were manually annotated for the tumour region, and a standardized ellipse representing an area of 1.07 mm^2^ inside the tumour was marked for cell counting. Cell detection was conducted using QuPath’s built-in ‘Positive cell detection’. For each immune cell subtype, the total number of tumour-infiltrating positive cells per mm^2^ was counted.

### 2.4. Statistical Analysis

Statistical analysis of data was performed using SPSS software (IBM Corp. Released 2019. IBM SPSS Statistics for Macintosh, Version 26.0. Armonk, NY, USA: IBM Corp). Pearson’s chi-squared test was applied to evaluate the correlation between clinico-pathological features. The correlation between infiltrating immune cell subtypes was calculated using Pearson correlation coefficient. The associations between tumour-infiltrating lymphocytes as well as PD-L1 expression and clinico-pathological parameters were analysed with the Kruskal-Wallis test. Outcomes of interest included recurrence-free survival, metastasis-free-survival and disease-specific survival. Kaplan–Meier survival analysis and log-rank test were used to analyse survival data and compare the distribution between groups. Univariable analyses were performed to assess the prognostic impact of clinic-pathological parameters on recurrence-free survival, metastasis-free-survival and disease-specific survival. Given the limited number of patients and outcome events of interest in this study, multivariable analyses were not performed. All tests were performed two-sided, and *p*-values ≤ 0.05 were considered statistically significant.

## 3. Results

### 3.1. Patient Characteristics

Thirteen of 36 (36%) patients were male and 23 (64%) patients were female, with a median age at diagnosis of 54 years (range: 20 to 91 years). Detailed clinico-pathological parameters are provided in Table 2. Primary tumour location was the parotid gland in 35 patients and the submandibular gland in the remaining patient. Tumour size ranged between 0.6 and 6 cm with a median size of 2.6 cm. Local recurrence occurred in four patients (11%), lymph node metastasis in six patients (17%) and distant metastases in seven patients (19%), with a median follow-up of 36 months (range: 5–168 months). Older patients were more likely to exhibit lymph node metastasis (*p* = 0.04), while female sex was associated with distant metastasis (*p* = 0.027). There was a significant correlation between tumour stage and metastasis (*p* = 0.007), as well as between high-grade transformation and lymph node metastasis (*p* < 0.001). Regional lymph node metastasis was significantly correlated with shorter metastasis-free survival (*p* = 0.006) and disease-specific survival (*p* < 0.001). There was a significant correlation between grading and metastasis-free survival (*p* = 0.001) as well as disease-specific survival (*p* = 0.016).

### 3.2. Quantification of TILs and PD-L1 Expression

The median number of CD3^+^ TILs was 721 cells per mm^2^ (range: 29–5456 cells). CD4^+^ cells were identified with a median number of 293 cells per mm^2^ (range: 5–5308 cells). The median number of tumour-infiltrating CD8^+^ cells was 276 cells per mm^2^ (range: 2–3507 cells). For CD20^+^ immune cell infiltration, the median number was 42 cells per mm^2^ (range: 0–2223 cells). There was a strong positive correlation between CD3 and CD4/CD8 positive immune cell counts (Pearson correlation coefficient >0.67), and an intermediate positive correlation between CD3 and CD20 positive immune cell counts (Pearson correlation coefficient = 0.58). Tertiary lymphoid structures were observed in eleven AciCCs (31%). There was a significant association between the number of tumour-infiltrating immune cells and the presence of tertiary lymphoid structures (*p* < 0.05; Table 3). The mean percentage of PD-L1 positive tumour cells (TPS) was 5.4% (median: 0%, range: 0–40%), the mean percentage of PD-L1 expression on tumour-infiltrating immune cells per tumour area (IC-Score) was 8.3% (median: 4.5%, range: 0–40%) and the mean CPS was 17.5 (median: 6.5, range: 0–80). There was a strong significant association between the number of TILs and PD-L1 expression (*p* < 0.05), with significant correlations for all combinations of immune cells and PD-L1 expression scores (e.g., CD3 positive immune cells vs. TPS: *p* < 0.001; Table 3). The presence of tertiary lymph structures was also significantly correlated with the PD-L1 expression scores IC-Score and CPS (*p* = 0.007 and *p* = 0.013, respectively), but not with the TPS score (*p* = 0.124).

### 3.3. Association between TILs, PD-L1 Expression and Clinico-Pathological Features

There was a significant association between the number of TILs and the presence of lymph node metastasis (Table 3, Figure 2A). The median number of T cells (CD3 positive immune cells) per mm^2^ was 277 in AciCCs with pN0, 1973 in pN1 and 3678 in pN2b, and therefore 7-fold higher in pN1 and 13-fold higher in pN2b compared to pN0 (Kruskal-Wallis test, *p* = 0.022). The difference for T cell subpopulations regarding CD4 and CD8 positive immune cells was also significant (*p* = 0.019 and *p* = 0.045, respectively) comparing AciCCs with pN0, pN1 and pN2b lymph node status, while there was no significant difference regarding B cells (CD20 positive immune cells; *p =* 0.189). Furthermore, the number of immune cells was significantly correlated with high-grade transformation (Table 3, Figure 2B). The median number of T cells (CD3 positive immune cells) per mm^2^ was 254 in conventional low-grade AciCCs compared to 2431 in AciCCs with high-grade transformation, and therefore almost 10-fold higher in high grade AciCCs (*p* < 0.001). There were also significantly higher counts of T cell subpopulations with CD4 expression (*p* < 0.001) or CD8 expression (*p =* 0.012), and also significantly higher counts for B cells (CD20 positive immune cells; *p =* 0.029) in high grade AciCCs. There were no significant associations between TILs and the parameters age, sex, tumour stage, resection status and clinical follow-up.

There was a significant correlation between increased PD-L1 expression on tumour cells (TPS, *p* = 0.015), as well as all PD-L1 staining cells (CPS, *p =* 0.028) and lymph node metastasis, while there was no significant difference for PD-L1 expression on immune cells (IC-Score). The mean CPS was 14.0 in AciCCs with pN0, 25.0 in patients with pN1 and 43.8 in pN2b, and therefore approximately 2-fold higher in pN1 and 3-fold higher in pN2b compared to pN0 (Figure 2C). Furthermore, there was a significant association between increased PD-L1 expression on tumour cells (TPS, *p =* 0.001), immune cells (IC-Score, *p =* 0.001), as well as all PD-L1 staining cells (CPS, *p* < 0.001) and high-grade transformation. The mean CPS was 10.7 in conventional low-grade AciCCs compared to 37.8 in AciCCs with high-grade transformation, and therefore almost 4-fold higher in high grade AciCCs (Figure 2D). Patients with higher age had a higher expression of PD-L1 on immune cells (IC-Score: *p =* 0.007; CPS: *p =* 0.11), while there were no further significant correlations between PD-L1 expression and other clinico-pathological features (Sex, tumour stage and resection status).

### 3.4. Association between Tils, Pd-L1 Expression and Clinico-Pathological Features with Clinical Follow-Up

The clinico-pathological features sex (*p =* 0.011) and tumour stage (*p =* 0.009) had a significant association with the metastasis-free survival, and the parameters lymph node status (*p =* 0.006 and *p =* 0.001), tumour grading (*p =* 0.001 and *p =* 0.016) as well as resection status (*p =* 0.011 and *p =* 0.023) were significantly associated with metastasis-free survival and disease-specific survival. The PD-L1 CPS score correlated significantly with the metastasis-free survival (*p =* 0.049, Figure 3), while the number of TILs and the PD-L1 expression scores TPS and IC-Score tended to be higher in the AciCC patients with less favourable follow-up, but this was not significant (*p* > 0.05).

## 4. Discussion

The prognostic value of TILs and PD-L1 expression in AciCCs have seldom been explored, although immune checkpoint inhibition is an emerging therapeutic strategy. In the present study, the number of T cell infiltrates in the tumour tissue was much higher compared to the number of B cells, which is in line with the general concept that T cells are the primary effectors of the immune system in cancer tissue [39,40,41]. Chen and Mellman described the anti-cancer immune response as the cancer-immunity cycle [42]. Cancer cell antigens are captured by dendritic cells and presented on MHCI and MHCII molecules to T cells [42]. Moreover, there are CD3+ subpopulations of macrophages, that have the ability to present antigens to other cells [43]. The recognition of cancer antigens results in the priming and activation of effector T cell responses in the lymph nodes. The activated effector T cells infiltrate the tumour and identify cancer cells through the interaction between T cell receptor and tumour cell antigen, which eventually initiates tumour cell killing [42]. B cells present the second most tumour-infiltrating lymphocytes [41] and play an important role in the tumour-associated immune cell response. On the one hand, B cells can serve as positive mediators of the antitumour response through the production of tumour-reactive antibodies, promoting tumour killing by natural killer cells, phagocytosis by macrophages, and the priming of CD4+ and CD8+ cells [41]. On the other hand, B cells can enhance tumour development through the production of autoantibodies, the expression of immunsuppressive ligands or cytokines that contribute to the inhibition of anti-tumour immune response [40,41]. Regulatory B cells negatively regulate immune responses by inhibiting Type 1 T helper cells and CD8+ cytolytic T cell responses [41]. In the current study, the number of T cells correlated with the presence of lymph node metastasis and with tumour grading, while the number of B cells correlated with tumour grading. Overall, this indicates an increased recognition of the tumour cells by TILs in advanced tumours with high grade transition, which are more likely to metastasize. One possible explanation for the association between increased immune cell infiltration and lymph node metastasis could be the expression of inhibitory co-receptors. Arolt et al. depicted the expression of lymphocyte activation gene 3 (LAG3) in 50% of AciCCs, which is expressed on TILs (mainly CD8+ and CD4+) and which was associated with advanced nodal metastases [44]. LAG3-positivity was associated with an inflamed tumour micro-environment and cytotoxic T cell infiltrate [44]. Furthermore, the collaboration of co-inhibitory receptors LAG3 and PD-L1 lead to inhibitory effects on T cell signalling [45]. High-grade transformation is associated with worse clinical outcome. A possible explanation for the association between high-grade transformation and increased infiltration of immune cells is a higher number of neoantigens in biological aggressive tumours that can be recognized by TILs. Linxweiler et al. reported that the level of immune cell infiltration in salivary gland carcinomas are associated with mutation- and fusion-derived neoantigens and with aggressive clinical behaviour [46]. Studies that investigated immune cell infiltration in other types of cancer also reported that increased immune cell infiltration is associated with a higher neoantigen load [47]. This correlates with the observed increased immune cell infiltration in AciCCs with high-grade transformation.

So far, not much is known about the role of PD-L1 in AciCCs. PD-L1 expression in AciCCs is rarely described as there are is no established criteria for PD-L1 scoring in AciCCs. Different antibodies and cut-offs have been applied, therefore the comparability between results is limited. In a study by Vital et al., where 1% cutoff for PD-L1 (clone SP142) positivity was applied, the authors identified 13% of PD-L1 positive AciCCs and 20% of tumours with PD-L1 positive infiltrating immune cells [29]. Witte et al. evaluated PD-L1 (clone SP263) expression in 16 AciCCs and described a median TPS of 1% and CPS of 3.5 in AciCCs, which was low compared to other salivary gland carcinomas PD-L1 [27]. The present study evaluating a cohort of 36 AciCCs observed comparable results of PD-L1 expression with PD-L1 28-8 clone. For the first time, our present study demonstrates an association between PD-L1 expression and lymph node metastasis as well as grading in AciCCs. These results are in line with other studies examining PD-L1 expression in mixed cohorts of salivary gland carcinomas. Witte et al. reported a significant association between CPS and lymph node metastasis in salivary gland carcinomas [27] and Vital et al. reported that PD-L1 expression in tumour cells and TILs in salivary gland carcinomas were related to higher tumour grading [29]. Higashino et al. found, that PD-L1 expression in parotid carcinomas was associated with a higher-stage, higher-grade and node positive cases [48]. Most studies report a negative impact of PD-L1 expression on outcome in salivary gland carcinomas [25,26], but there is no study on the prognostic impact of PD-L1 expression in AciCCs. In the present study, there was a significant correlation between an increased PD-L1 expression (CPS) and shorter metastasis-free survival for the first time. This association warrants further studies, especially in the context of anti-PD-1/PD-L1 inhibitors as emerging treatment options. The PD-1/PD-L1 pathway has evolved as a key immune checkpoint. While PD-1 represents a coinhibitory molecule which is predominantly presented on tumour specific T cells, PD-L1 is expressed on the surface of antigen presenting cells such as natural killer cells, dendritic cells, lymphocytes and tumour cells [49]. PD-L1 expressed on the tumour cells binds to PD-1 receptors on activated T cells, blocking the cytotoxic activity of the effector T cells. Upregulation of PD-L1 has been demonstrated in salivary gland cancers and results of recently published studies indicate a possible use of checkpoint inhibitors. Pembrolizumab treatment in advanced salivary gland carcinomas was associated with an overall response rate of 12% [30] and 16 % [31], respectively. Rodriguez et al. reported a partial response to Pembrolizumab in combination with Vorinostat in two of three patients with AciCC [31]. However, the PD-L1 expression in these three tumours was not mentioned in that study. Noteworthy, within the cohort of 18 patients with salivary gland cancer in that study, two out of only four patients that showed a partial response had an AciCC [31]. Generally, PD-1/PD-L1 blockage correlates with PD-L1 expression, although not in every single patient [50]. Furthermore, response rates to anti PD-1/PD-L1 are generally higher in tumour types with higher mutational burdens [51,52]. A recent study employing whole genome sequencing observed that the tumour mutational load (TMB) as well as structural variations in AciCC were rather low compared to other cancer entities, but that study included mostly low grade AciCCs [53]. In summary, to date the available data on patients with AciCC undergoing anti-PD-1/PD-L1 inhibitor therapy is sparse. However, the current study demonstrating a significant increase in TILs in correlation with high grade advanced AciCCs warrants the evaluation of the predictive value of PD-L1 expression as well as further biomarkers like TMB with regard to the efficacy of immunotherapies.

Regarding the immune escape from PD-L1/PD-1 targeted therapy, no specific data on AciCC exists to date. In principle, Kim and Chen described several potential factors that had been linked to immune escape: The lack of cancer antigens or epitopes recognized by T cells, reduced T cell infiltration or rather insufficient activation of effector T cells, downregulation of the MHC molecules on cancer cells and the presence of an immunsuppressive tumour microenvironment [54]. Additionally, different mechanisms of intratumoural T cell suppression may lead to checkpoint blockade failure [52]. Therefore, alternative checkpoint inhibitors, such as lymphocyte activation gene-3 (LAG3), T cell immunoglobulin-mucin domain 3 (TIM3), T Cell Immunoglobulin and ITIM domain (TIGIT), as well as indoleamine 2,3-dioxygenase (IDO) or transforming growth-factor-β (TGF-β) [52], should be considered as a treatment option as well. Although a subset of ~17.9% of AciCCs has been reported to express EGFR [55], only few clinical trials regarding anti-EGFR therapy in salivary gland carcinomas have been performed and the treatment did not result in significant clinical responses so far [55,56]. Combination therapies offer an opportunity to create a highly active anti-tumour microenvironment in which coinhibitory pathway blockade can amplify and broaden anti-tumour immune response [52]. Especially the combination of anti-LAG3 and anti-PD-L1 has shown efficacy against PD-L1 resistance [57]. In conclusion, increased immune cell infiltration of T and B cells as well as higher levels of PD-L1 expression in AciCCs in association with high-grade transformation, lymph node metastasis and unfavourable prognosis suggests a relevant interaction between tumour cells and immune cell infiltrates and warrants further exploration of the efficacy of immunotherapies in clinical studies with inclusion of high grade AciCCs only.

## 5. Conclusions

Taken together, our study shows for the first time that an increased number of TILs and higher PD-L1 expression levels correlate with the unfavourable clinico-pathological parameters lymph node metastasis and high-grade transformation in AciCC. This association likely reflects an improved recognition of the cancer cells by the immune system paralleling tumour evolution and dedifferentiation. Regarding the potential benefit of patients with AciCC from immune checkpoint blockade inhibitors, our study clearly provides a rationale to initiate further prospective clinical trials focusing on AciCCs with high-grade transformation.

## Figures and Tables

**Figure 1 cancers-13-00965-f001:**
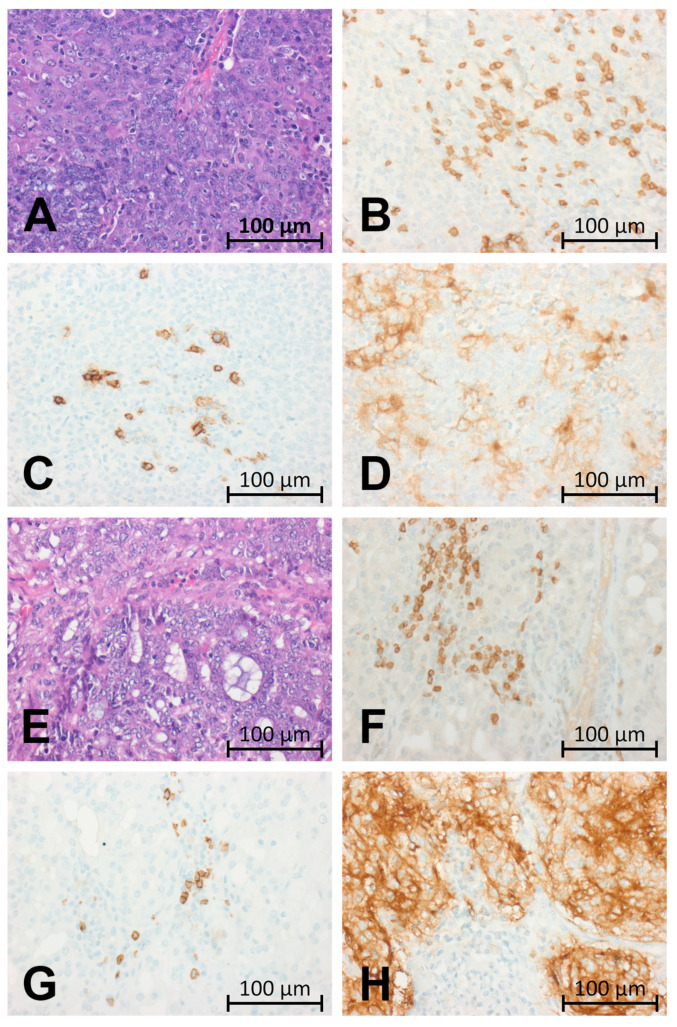
Representative examples of two AciCCs with high-grade morphology ((**A**–**D**), case 6, (**E**–**H**), case 35). (**A**) H&E staining (×400) of case 6 shows high-grade dedifferentiation. (**B**) Anti-CD3 staining (×400) reveals a high number of intra-tumoural T cells. (**C**) Anti-CD20 (×400) staining shows increased intra-tumoural B cells. (**D**) Anti-PD-L1 staining (clone 28-8, ×400) is highly expressed in the tumour cells. (**E**) H&E staining (×400) of case 35 with high-grade dedifferentiation. (**F**) Anti-CD3 staining (×400) reveals a high number of intra-tumoural T cells. (**G**) Anti-CD20 (×400) staining shows increased intra-tumoural B cells. (**H**) Anti-PD-L1 staining (clone 28-8, ×400) is highly expressed in the tumour cells.

**Figure 2 cancers-13-00965-f002:**
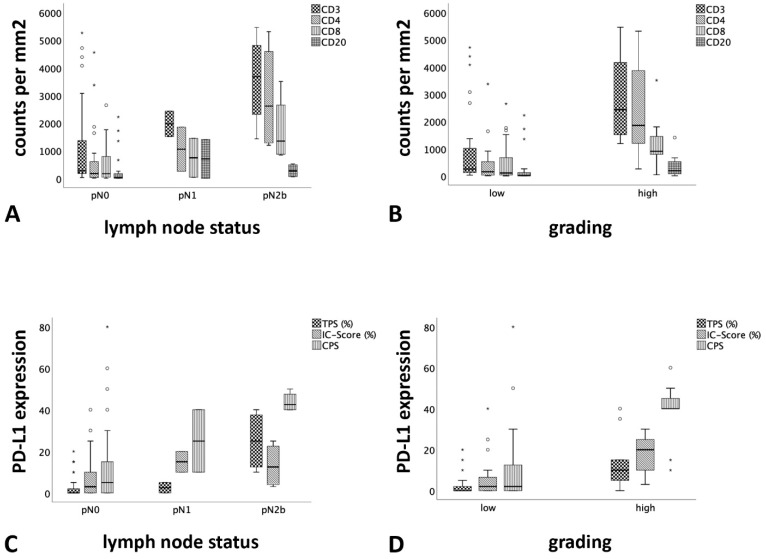
Association between TILs and PD-L1 expression with lymph node status and grading. (**A**) correlation between lymph node status and TILs (CD3: *p =* 0.022; CD4: *p =* 0.019; CD8: *p =* 0.045; CD20: *p =* 0.189). (**B**) correlation between grading and TILs (CD3: *p* < 0,001; CD4: *p* < 0,001; CD8: *p =* 0.012; CD20: *p =* 0.029). (**C**) correlation between lymph node status and PD-L1 expression (TPS: *p =* 0.015; IC-Score: *p =* 0.160; CPS: *p =* 0.028). (**D**) correlation between grading and PD-L1 expression (TPS: *p =* 0.001; IC-Score: *p =* 0.001; CPS: *p* < 0.001). The lymph node status and grading are indicated on the x-axis, while the counts of the respective TILs per mm^2^ and the PD-L1 expression are represented on the y-axis. Horizontal lines on box-plots delineate quartiles, with outlier and extreme values indicated by circles (◦) and asterixes (*), respectively. CPS, combined positive score; IC-Score, immune cell score; TPS, tumour proportion score.

**Figure 3 cancers-13-00965-f003:**
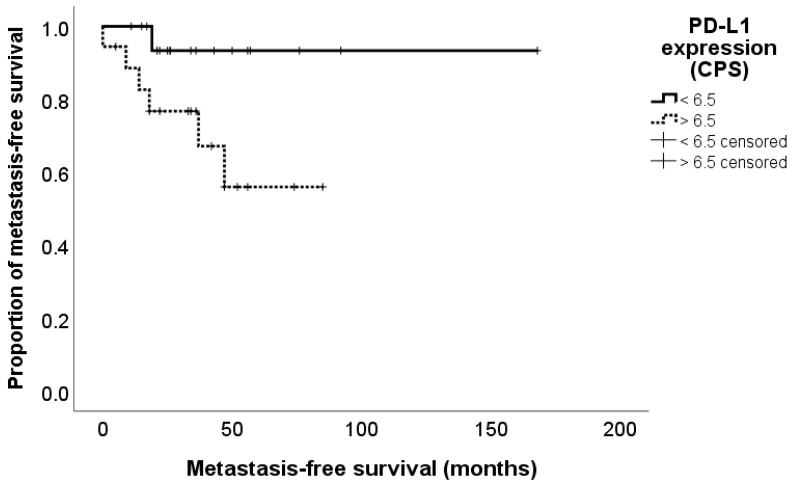
Correlation between PD-L1 expression (CPS) and metastasis-free survival. Shown is the Kaplan-Meier estimate for metastasis-free survival of 36 AciCCs with low (<6.5, solid line) and high (>6.5, dashed line) levels of PD-L1 expression (CPS). The statistical difference is significant (*p =* 0.049). The x-axis shows the metastasis-free survival in months. CPS, combined positive score.

**Table 1 cancers-13-00965-t001:** Antibodies and staining conditions.

Antibody	Source	Clone	Dilution	Incubation Time
CD3	Zytomed	SP7	1:150	32 min
CD4	Quartett	EP204	1:100	60 min
CD8	Dako	CD8/144B	1:100	32 min
CD20	Dako	L26	1:100	20 min
PD-L1	abcam	28-8	1:200	60 min

**Table 2 cancers-13-00965-t002:** Clinico-pathological data of 36 primary AciCCs.

pat. id	Sex/Age	Location	Size (cm)	pT/pN/R	Grading	Follow-Up (ms)	pfs/mfs/dss/dot
1	m/50	smg	0.6	1/0/0	low	rec. (63)/awd (168)	63/168/168/no
2	f/34	pg	0.7	1/0/0	low	ned (21)	21/21/21/no
3	m/30	pg	1.2	1/0/0	low	ned (85)	85/85/85/no
4	m/31	pg	1.2	1/0/0	low	ned (76)	76/76/76/no
5	f/24	pg	1.2	1/0/0	low	ned (17)	17/17/17/no
6	f/48	pg	1.5	1/2b/0	high	met. (9)/dot (18)	9/9/18/yes
7	f/60	pg	1.5	1/0/0	low	ned (42)	42/42/42/no
8	f/32	pg	1.5	1/0/0	low	ned (5)	5/5/5/no
9	m/59	pg	1.7	1/0/0	high	ned (74)	74/74/74/no
10	m/87	pg	1.7	1/X/1	low	rec. (38)/dooc (43)	38/43/43/no
11	f/49	pg	1.7	1/0/0	low	ned (26)	26/26/26/no
12	f/22	pg	1.8	1/0/0	low	ned (57)	57/57/57/no
13	f/55	pg	1.8	1/0/0	low	ned (34)	34/34/34/no
14	m/74	pg	1.9	1/0/0	low	dooc (36)	36/36/36/no
15	m/53	pg	2.0	1/0/0	low	ned (15)	15/15/15/no
16	f/35	pg	2.1	2/0/0	low	ned (50)	50/50/50/no
17	f/36	pg	2.3	2/0/0	low	ned (22)	22/22/22/no
18	m/68	pg	2.5	2/0/0	low	ned (56)	56/56/56/no
19	m/59	pg	2.6	2/0/0	low	ned (33)	33/33/33/no
20	m/20	pg	2.8	2/0/0	low	ned (56)	56/56/56/no
21	f/43	pg	2.8	2/0/0	high	ned (36)	36/36/36/no
22	f/77	pg	2.9	2/1/0	high	met. (18)/awd (22)	18/18/22/no
23	f/42	pg	3.0	2/0/0	low	ned (11)	11/11/11/no
24	f/40	pg	3.0	2/0/0	low	ned (26)	26/26/26/no
25	f/91	pg	3.2	3/0/0	low	dooc (18)	18/18/18/no
26	m/65	pg	3.4	3/1/0	low	ned (22)	22/22/22/no
27	f/65	pg	3.5	3/0/1	high	met. (19)/awd (38)	19/19/38/no
28	m/77	pg	3.5	2/0/0	low	dooc (47)	47/47/47/no
29	f/68	pg	3.5	4a/0/0	low	ned (34)	34/34/34/no
30	f/78	pg	3.9	2/2b/0	high	ned (52)	52/52/52/no
31	f/34	pg	4.0	2/0/0	low	ned (25)	25/25/25/no
32	f/26	pg	4.5	3/0/0	high	rec. (3), met. (14), awd (60)	3/14/60/no
33	f/74	pg	4.5	3/2b/0	high	met. (37), awd (44)	37/37/44/no
34	m/56	pg	4.5	3/0/0	low	rec. (28), awd (92)	28/92/92/no
35	f/70	pg	5.8	3/2b/1	high	met. (0), dot (6)	0/0/6/yes
36	f/72	pg	6.0	3/0/0	low	met. (47), awd (53)	47/47/53/no

awd, alive with disease; dooc, died of other cause; dot, died of tumour; dss, disease specific survival; f, female; pg, parotid gland; m, male; mfs, metastasis-free survival; ms, months; ned, no evidence of disease; pfs, progression-free survival, pN, pathological N-classification; pT pathological T-classification; R, resection status; rec., recurrence; smg, submandibular gland. The exact criteria according to the current TNM classification [38] for the different stages of primary tumour (pT) are: pT1, Tumour ≤2 cm in greatest dimension, without extraparenchymal extension; pT2, Tumour > 2 cm but ≤ 4 cm in greatest dimension, without extraparenchymal extension; pT3: Tumour > 4 cm and/or with extraparenchymal extension; pT4a: Tumour invades skin, mandible, ear canal, or facial nerve. The pathological N-classification describes the regional lymph nodes (pN) (i.e., the cervical nodes) as follows: pNX, Regional lymph nodes cannot be assessed; pN0, No regional lymph node metastasis; pN1, Metastasis in a single ipsilateral lymph node, ≤ 3 cm in greatest dimension; pN2b, Metastasis in multiple ipsilateral lymph nodes, all ≤ 6 cm in greatest dimension.

**Table 3 cancers-13-00965-t003:** Correlation between immune cell infiltration as well as PD-L1 expression and clinico-pathological features (*p*-values).

Attribute	TLS	CD3	CD4	CD8	CD20	TPS	IC-Score	CPS	PFS	MFS	DSS
age	0.660	0.323	0.126	0,171	0.963	0.152	**0.007**	**0.011**	0.937	0.351	0.929
sex	0.127	0.239	1.000	0.871	0.948	0.312	0.820	0.871	0.585	**0.011**	0.269
pT	0.271	0.523	0.613	0.526	0.444	0.639	0.295	0.324	0.066	**0.009**	0.689
pN	0.819	**0.022**	**0.019**	**0.045**	0.189	**0.015**	0.160	**0.028**	0.127	**0.006**	**0.001**
grading	0.621	**<0.001**	**<0.001**	**0.012**	**0.029**	**0.001**	**0.001**	**<0.001**	0.895	**0.001**	**0.016**
R	0.230	0.114	0.377	0.199	0.347	1.000	0.442	0.829	0.192	**0.011**	**0.023**
PFS	0.505	0.954	0.998	0.901	0.958	0.587	0.290	0.290	-	-	-
MFS	0.202	0.244	0.205	0.318	0.259	0.100	0.257	**0.049**	-	-	-
DSS	0.316	0.170	0.145	0.170	0.170	0.121	0.998	0.158	-	-	-
TPS	0.124	**<0.001**	**<0.001**	**<0.001**	**0.007**	-	-	-	-	-	-
IC-Score	**0.007**	**0.003**	**<0.001**	**0.006**	0.214	-	-	-	-	-	-
CPS	**0.013**	**0.001**	**<0.001**	**0.001**	0.143	-	-	-	-	-	-
TLS	-	**0.002**	**<0.001**	**<0.001**	**<0.001**	0.124	**0.007**	**0.013**	-	-	-

CPS, combined positive score; DSS: disease-specific survival; IC-Score, immune cell score; MFS, metastasis-free survival; pfs, progression-free survival; pN, pathological N-classification; pT, pathological T-classification; R, resection status; TLS, tertiary lymphoid structures; TPS, tumour proportion score. Significant *p*-values (<0.05) are indicated in bold.

## Data Availability

The data presented in this study are available on request from the corresponding author. The data are not publicly available due to ethical reasons.

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
