# Peer review of "Tumour-Infiltrating Lymphocytes (TILs) and PD-L1 Expression Correlate with Lymph Node Metastasis, High-Grade Transformation and Shorter Metastasis-Free Survival in Patients with Acinic Cell Carcinoma (AciCC) of the Salivary Glands"

_cancers, 2021, doi:10.3390/cancers13050965_

Round 1

Reviewer 1 Report

General comments

In this paper the authors evaluate the number of tumor-infiltrating lymphocytes and the extent of PD-L1 expression in a limited number of Acinic Cell Carcinoma of the salivary glands. They try to correlate these values with clinico-pathological features and to analyze their prognostic impact.

The paper is well written and the results are convincing. The findings show that a high level of lymphocytes infiltration correlate with high grade transformation and node positive disease. Furthermore a significant correlation was found between higher level of PD-L1 expression, lymph node metastasis , high–grade transformation and unfavourable prognosis.

While the recruitment of lymphocytes and the PD-L1 expression are important features, also the recruitment of macrophages in the tumour may be relevant. The possibility that the number of these cells could play a role should be mentioned in the discussion. Of note activated macrophages can express CD3 (doi.org/103389/fimmu.2019.02550)

Specific comments

Line 127: the authors correctly state that …”given the limited number of patients…..”. Accordingly the sentence (line 236) ..”evaluating a relative large cohort…” should be substituted with …” a cohort…”

Table 2: a brief description of the meaning of PN0, PN1, PN2/2b and of PT 1-3 is required for those who are not familiar with this type of tumour

The possible involvement of B cells is not clear and deserves to be better described. IC-Score and CPS are Non Significant. TPS is significant, but, as shown in Fig.1, the counts per mm2 are very low even in PN2 nodes and in high grade ca with respect to T cells. Are antibody production active against AciCC tumour antigens?

Fig.1 : the findings reported in this figure should be described in the Result section on the basis of PN0-2b classification. This part is missing. The p values should be indicated also in the legend. 

Legend to Fig.2: panel C is missing: intratumoral T lymphocytes…..C, in contrast---; D, Anti-PD-L1 and not PL-L1

Line 192: a value of 0.0% for a range of 0-40% is unlikely: please correct!

Fig. 3 abscissa must be quoted

Reviewer 2 Report

Dear Author,

Thank you for submitting a manuscript titled “Tumour-infiltrating lymphocytes (TILs) and PD-L1 expression correlate with lymph node metastasis, high-grade transformation and shorter metastasis-free survival in patients with acinic cell carcinoma (AciCC) of the salivary glands”. Authors report valuable and useful information for treatment of AciCC. It clearly shows the relationship between TILs/PD-L1 expression and AciCC prognosis. Before accepting, however, there are some concerns listed up below.

Major concern:

  1. There is only one case shown in Figure 2. It is more informative visually and favorable if histological evidences from multiple cases are shown, using data such as case 6 or/and case 35.

Minor concerns:

  1. It is favorable to explain the definition of and how to calculate Tumor Proportion Score (TPS), Ventana immune cell score (IC-Score) and Combined Positive Score (CPS).
  2. In figure 2 legend, it looks there is no description about Figure 2C.
  3. In figure 2, it is favorable to show scale bars.
  4. On line 192, authors describe TPS was 0.0% (range 0-40%). Does this mean led than 0.05%? It is better to describe the exact number such as 0.028%.

Reviewer 3 Report

The manuscript needs major revisions.

Material and methods
2.2. The description of the methodology should be detailed.
2.4. How was the number of patients determined?

Results
From a clinical point of view, correlation analysis is insufficient. I propose to use stepwise multivariable linear regression model to analyze the dependence of every parameter by explicable variables as gender, age, BMI, etc.
Description of results needs improvement. Correlations between data should be distinguished from differences between groups.
Figure 1: I suggest to calculate p-trend quartiles of measured parameters and compare with appropriate test. Provide p-values on the graphs.

Discussion
The discussion is laconic. It does not discuss the issue presented. The authors should attempt a mechanistic view of the topic. Moreover, what is the clinical significance of the results?
What is the next step? What are the limitations of the study?

Round 2

Reviewer 3 Report

Publish